# Effectiveness of adding inspiratory muscle training to a cardiac rehabilitation program in people with acute myocardial infarction revascularized by percutaneous coronary intervention (CARDIOINSPIRE): Study protocol for a randomized controlled trial

Jose M. Zuazagoitia-Lama-Noriega[1]*, A. M. Gómez-González[2,3], Jose A. Moral-Munoz[1,4]

1 Department of Nursing and Physiotherapy, University of Cadiz, Cadiz, Spain, 2 Rehabilitation Department, Virgen de la Victoria University Hospital, Malaga, Spain, 3 Biomedical Research and Innovation Institute of Malaga and Nanomedicine Platform (IBIMA-BIONAND Platform), Malaga, Spain, 4 Biomedical Research and Innovation Institute of Cadiz (INiBICA), University of Cadiz, Cadiz, Spain

* josemaria.zuazagoitia@uca.es

## Abstract

### Objectives

This study aims to analyze the effectiveness of adding inspiratory muscle training (IMT) to a cardiac rehabilitation program (CRP) in people with acute myocardial infarction (AMI) revascularized by percutaneous coronary intervention (PCI) after 16 sessions. The biopsychosocial profile and the sex differences of this population will also be evaluated.

### Design

Triple-blind, parallel-group, low-risk randomized controlled trial.

### Methods

72 patients diagnosed with AMI will be enrolled and randomly assigned to two groups. The control group will complete the usual CRP with the addition of IMT at 5% of the maximal inspiratory pressure (MIP) (sham load). The intervention group will perform the same CRP but will add an IMT program at 70% of MIP. Outcomes will be collected at baseline and post-intervention. The main outcome will be cardio-respiratory fitness (CF) measured in metabolic equivalent of task (MET), secondary outcomes will be MIP, maximal expiratory pressure, peripheral muscle strength, dyspnea, social support, anxiety, depression, coping with the disease, sexual dysfunction, quality of life (QoL), sleep quality, eating habits, and body composition. Descriptive statistics will summarize baseline data, and influential outliers will

**Data availability statement:** No datasets were generated or analysed during the current study. All relevant data from this study will be made available upon study completion.

**Funding:** This study was supported by the Spanish Society of Cardiac and Respiratory Rehabilitation (Sociedad Española de Rehabilitación Cardiaca - SORECAR) through the 2024 Best Project Award, grant number BEC-SOR-PI24/01, awarded to authors J.M.Z-L.-N. and A.M.G-G.(https://www.sorecar.net). Additionally, authors J.M.Z-L.-N. and J.A.M-M. received funding from the Distinguished Professional Association of Physiotherapists of Andalusia (Ilustre Colegio Profesional Fisioterapeutas de Andalucía - ICPFA) under the 2025 Research Grant Program (grant number AI-15/2026) (https://www.colfisio.org). The funders had no role in study design, data collection and analysis, decision to publish, or preparation of the manuscript".

**Competing interests:** The authors have declared that no competing interests exist.

be managed via winsorization or exclusion, confirmed by sensitivity analyses. A mixed-design ANOVA will evaluate differential changes over time. Depending on attrition rates, missing data will be addressed using either complete case analysis (<5%) or multiple imputation with an intention-to-treat approach (>5%).

## Discussion

According to recent literature, improvements in CF and MIP are expected. For the rest of the outcomes, data are currently limited, inconclusive or lacking. We hypothesize that changes in CF may affect the rest of the outcomes and, in our opinion, IMT at 70% of the MIP will significantly improve CF.

## Clinical implications

If the expected benefits are observed, the results may recommend including IMT as a fundamental component of a CRP in our study population.

## Trial registration number

Registered prospectively at ClinicalTrials.gov (NCT06681740) Version 4.

## Introduction

Acute myocardial infarction (AMI) is characterized by narrowing of the internal lumen of the arteries that perfuse the heart. As the arterial lumen narrows, blood flow decreases, consequently diminishing the oxygen supply to the myocardium. This condition may lead to angina pectoris or may potentially result in acute myocardial infarction [1–3]. It is the leading cause of death from cardiovascular disease and is responsible for 9.44 million deaths in 2021 and 185 million disability-adjusted life years [4]. Therefore, it is one of the most important public health problems worldwide, and entail a high health and social spending [5]. In that way, percutaneous coronary intervention (PCI) is one of the most common treatments for AMI because it is a minimally invasive technique [6]. In addition to this surgical technique to revascularize the myocardium, it is necessary to prevent further cardiovascular events. Cardiac rehabilitation programs (CRPs) are widely recognized as effective non-pharmacological interventions for improving the quality of life (QoL) and preventing subsequent cardiovascular events in patients with AMI [7–10]. For these and other reasons, CRPs are recommended as an essential intervention after coronary events in the clinical practice guidelines of major international associations such as the European Society of Cardiology [11], the American Heart Association and the American Association of Cardiovascular and Pulmonary Rehabilitation [12]. These include smoking cessation, control of other cardiovascular risk factors, health education, psychological therapy, and multimodal exercise programs [13–16]. Two types of exercise are usually performed: endurance and strength training [12]. The third type of exercise is inspiratory muscle training (IMT), which is widely used in patients with heart failure (HF) [17].

This involves applying higher-than-usual workloads to the muscles responsible for generating the necessary negative pressure inside the thorax for air to enter the lungs. This overload induces adaptations that increase the muscle strength. It is a safe intervention that reduces dyspnea and improves cardiorespiratory fitness (CF) and QoL in these patients [18]. In that sense, a recent systematic review with meta-analysis by Fabero-Garrido et al. [16] concluded that the quality of the evidence for the use of IMT to improve maximal oxygen consumption and expiratory muscle strength in people who suffer from ischemic heart disease is very low. Therefore, the available literature analyzing the effectiveness of IMT in people with AMI who underwent revascularization by PCI is limited. Consequently, it is pertinent to conduct a low-risk randomized clinical trial to evaluate the effectiveness of adding IMT to a CRP in this specific group.

The primary aim of the proposed study is to analyze the effectiveness of adding IMT to a CRP in people with AMI who underwent revascularization by PCI after 16 intervention sessions (2 times/week), based on CF, maximal inspiratory pressure (MIP), maximal expiratory pressure (MEP), peripheral muscle strength, social support, anxiety, depression, coping with the disease, sexual dysfunction, QoL, sleep quality, eating habits, and body composition.

As a secondary aim, we will determine the biopsychosocial profile and analyze sex differences in this population through subgroup analyses.

## Materials and methods

### Design

An 8-week, triple-blind, parallel-group, low-risk, randomized controlled trial (RCT) will be conducted in patients with AMI who underwent revascularization by PCI. This protocol was reported according to the Standard Protocol Items: Recommendations for Interventional Trials (SPIRIT 2013 Statements) [19]. The checklist is available in the supporting information (S1 File). Furthermore, the Consensus on Exercise Reporting Template (CERT) has been included as Supporting Information (S6 File) to provide comprehensive rehabilitation details and guarantee the intervention's replicability.

### Participants and the study setting

This study will include patients with AMI who underwent revascularization by PCI. These individuals will be recruited from among patients admitted to a CRP in the Cardiac Rehabilitation Unit (CRU) of the Hospital Universitario Virgen de la Victoria in Málaga (HUVV), Spain. Physicians in charge of potential participants with AMI who underwent revascularization by PCI will propose inclusion in the study by applying the eligibility criteria. The recruitment began on November 11, 2024, and will end on December 31, 2025. Patients will receive general information and written consent forms from their doctors, who will ask them about their interest in participating in the RCT. If the answer is yes, they will be asked to sign the written informed consent form. This will be kept in a safe place by the researchers.

### Eligibility criteria

The inclusion criteria will be as follows: a) male and female patients with AMI who underwent emergency revascularization by PCI aged over 18 and under 80 years, and b) with absence of cognitive deficits and/or physical limitations that would prevent them from performing physical exercise or completing the questionnaires necessary for participation in the study. Patients will be excluded if they have undergone revascularization via coronary artery bypass grafting or if they present with one of the main absolute contraindications for initiating cardiac rehabilitation in patients with AMI: unstable angina, decompensated heart failure, severe uncontrolled arrhythmias, severe uncontrolled hypertension, and acute myocarditis or pericarditis [20].

### Interventions

Both study groups will undergo 16 intervention sessions (2 times/week) of a CRP, consisting of multicomponent exercise (endurance and strength training), health education, and weekly group psychological therapy sessions. This program

includes the core components of CRPs recommended by the American Heart Association and the American Association of Cardiovascular and Pulmonary Rehabilitation [12].

The CRP will be performed and supervised by the usual multidisciplinary team of the HUVV. The team is composed of a cardiologist, rehabilitation physician, endocrinologist, physiotherapist, nurse, and psychologist. The study participants will attend the CRU of the hospital three times per week. Two of these days will be dedicated to physical exercise sessions, and the third will be dedicated to health education and group psychological therapy sessions. The health education session will deal with a different topic each week (basic knowledge of their pathology, proper medication, safe physical exercise, sexual dysfunction, healthy eating, and control of cardiovascular risk factors). Subjects suffering from erectile dysfunction will be offered the possibility of being referred to a urologist. Participants with poorly controlled diabetes and/or morbid obesity will be referred to an endocrinologist on the CRU.

The training sessions will take place in the cardiac rehabilitation room at the HUVV in groups of 8 patients and will last approximately 75 minutes. This space has a rectangular floor plan, measures approximately 100m$^2$, and is air-conditioned. It contains the equipment described in Table 1 for conducting exercise sessions, as well as telemetry equipment for monitoring participants during exercise. In addition, it has everything necessary to respond in the event of any adverse effects or cardiopulmonary arrest. This includes a stretcher, wheelchair, sphygmomanometer, stethoscope, electrocardiograph, and a fully equipped crash cart with a defibrillator.

**Table 1. Structure of the cardiac rehabilitation exercise session.**

| Category | Description |
|---|---|
| Patient reception | Collection of constants (blood pressure, heart rate, and glycemia in case of diabetic patients), and placement of telemetry. |
| Warm-up | A battery of different full body exercises will be performed including diaphragmatic breathing, balance exercises, low load strength exercises and gentle endurance exercises. |
| Strength Exercise | It will consist of different exercises with dumbbells and elastic bands involving various joints. Participants will work at 50% of their 1 repetition maximum (1RM), measured indirectly using the 20 repetitions maximum (20RM) method. Three sets of 10 repetitions of the following exercises will be completed: elbow flexion, shoulder abduction and knee extension. The rest interval between sets will be 30 seconds. |
| Endurance Exercise | It will be performed on a treadmill or stationary bicycle. The duration of the training will be 30 minutes. In the first month, each patient will be trained at 70% of their heart rate resulting from applying the Karvonen formula, and in the second month, at 80%. The modality may be continuous or intervallic, depending on the patient's tolerance. |
| Relaxation and calm-down | Patients will remain seated in chairs for 5 min, during which a guided relaxation audio will be played in order to facilitate the return of their heart rates and blood pressures to baseline values |
| End of session and farewell | Vital signs will be checked to ensure that participants are leaving in appropriate condition. They will be given advice on healthy lifestyle habits before their departure. |

The training sessions will be led by a physiotherapist with previous experience and training in supervising exercise for people with heart disease. This professional will keep an attendance record on the computer in order to measure and report adherence. In addition, this physiotherapist will encourage patients in each session to continue with the program by reinforcing its benefits. The structure and other details of the sessions are shown in Table 1.

The program will be extended for eight weeks until 16 concurrent training sessions are completed.

In addition to the classic CRP described above, the intervention group will undergo IMT during the same period. The Power Breath model – Medic Classic device will be used. The IMT will be performed by adjusting the device load to 70% of the MIP value measured prior to the start of the program. Participants will complete one daily training session four days a week (Monday, Wednesday, Friday, and Sunday), three sets of 10 repetitions, and a rest of 3-minutes between sets. The MIP value will be measured again halfway through the program to readjust the training load. Previous studies with patients with coronary artery disease have used training protocols that are similar in terms of workload, number of repetitions and sets, and distribution of weekly sessions. These clinical trials reported no adverse events, supporting the safety and tolerability of high-intensity IMT in this study population [22–24]. We hypothesize that high-intensity IMT training may lead to greater inspiratory muscle adaptations, resulting in improved CF and other outcomes compared to lower-intensity training [21–23].

The control group will be given the same device as the experimental group, but the load for all participants in the control group will be 5 cm of $H_2O$. This load is so low that it cannot induce improvements in the inspiratory musculature. Therefore, it will act as a sham intervention in the control group.

A rehabilitation physician, with prior expertise and training in the practice of the device, will teach participants how to use it on the first day, and they will be instructed to do it individually at home. In order to prevent non-compliance with the IMT by the participants, they will be given a paper log to be filled in daily after the IMT session. At any time, participants may ask the rehabilitation physician who instructed them any questions they may have about using the device. During regular CRP exercise sessions, the physiotherapist will emphasize the importance of completing the prescribed IMT.

## Outcomes

To analyze the effectiveness of IMT, several outcomes will be assessed. CF as main outcome, PIM, PEM, peripheral muscle strength, social support, anxiety, depression, coping with the disease, sexual dysfunction, QoL, sleep quality, eating habits, and body composition as secondary outcomes. The following sociodemographic and clinical variables will also be collected to characterize the sample and explore potential associations with the outcomes of interest: age, biological sex, marital status, educational level, profession, and nationality.

In the CRU consultation, an anamnesis will be performed in which the following data will be collected for the clinical history: cardiac comorbidities (revascularization, HF, heart valve surgery), presence of cardiovascular risk factors (current or previous smoking, arterial hypertension, dyslipidemia, diabetes mellitus, sedentary lifestyle, obstructive sleep apnea), locomotor, neurological, vascular, or respiratory comorbidities, physical exercise habits (type and time), and stratification of the patient's cardiac risk (low, medium, or high).

**Sociodemographic outcomes.** Age will be recorded as a continuous variable in years, based on the participant's age at the time of data collection. Biological sex will be categorized as either male or female, based on the sex assigned at birth and recorded in official identification documents. Marital status will be classified into four categories: single, married, separated/divorced, and widowed. This variable reflects the participant's current relationship and legal status. Educational level will be assessed based on the highest level of formal education completed by each participant. Categories included: no formal education, mandatory basic education, high school diploma or vocational training, university degree programs (bachelor's degree, diploma) and postgraduate studies (master's or doctorate). Nationality will be defined as the country of legal citizenship at the time of the study. This variable was used to assess the cultural and legal contexts of each participant.

Occupation will be categorized according to the participant's current or most recent employment status and type of work, based on standard occupational classifications [24]. This included military; business and public administration management; technicians, scientists, and intellectuals; technical and support professionals; administrative employees; workers in catering, personal services, security and retail sales; skilled workers in agriculture and fisheries; craftsmen and skilled workers in manufacturing, construction, and mining, except plant and machine operators; skilled workers in the extractive industries, metallurgy, and construction; operators and assemblers of fixed installations and machinery; drivers and operators of mobile machinery; and unskilled workers.

**Clinical outcomes.** Comorbidities will be recorded at baseline based on medically documented diagnoses of chronic conditions coexisting with primary health concerns. Before and after the intervention, the following assessments and evaluations will be performed.

To assess CF, simple ergometry will be performed following the clinical guidance by consensus of recommendations for clinical exercise tolerance testing according to The Professional Body for Cardiac Scientists [25], in which the maximum oxygen consumption ($VO_2max$), duration of the test, and type of response will be measured. Ergometry, also known as a stress test, is a diagnostic technique that analyzes the response of the heart to exercise and quantifies the oxygen consumption of the individual performing the test. In this study, the exercise will be performed on a treadmill. The Bruce protocol, which progressively increases the speed of the treadmill and its inclination every 3 min, will be used. This procedure will be performed by a cardiologist in our unit. $VO_2max$ will be obtained in METS (a metabolic equivalent of task, 1 MET = 3.5 ml $O_2$/kg/min). The time and type of response will be recorded. This may be clinically positive if the cardiologist appreciates the appearance of angina on exertion or is electrically positive if a horizontal or downward depression of the ST segment of 1 mm measured at 80 ms from the J point is present.

Left ventricular ejection fraction (LVEF) will be measured by a cardiologist using ultrasound, following the recommendations of the American Society of Echocardiography for use in clinical trials and in agreement with expert cardiologists [26].

The total number of sessions attended by each participant will be recorded at the end of the intervention period.

**Strength measurements.** A MicroRPM® digital portable manometer (Vyaire Medical GmbH, Hoechberg, Germany) will be used to measure the MIP and MEP. These assessments will be performed according to the recommendations of the American Thoracic Society (ATS) [27] and the European Respiratory Society (ERS) [28] and following the protocol of the Spanish Society of Pneumology and Thoracic Surgery (SEPAR) [29]. MIP will be reassessed midway through the program (at the end of week 4) to readjust the training load.

Perypheral musculature will be assessed as follows. Maximum static quadriceps muscle strength will be measured with the patient seated on a stretcher and the knee flexed at 90° using a Hogan Health Industries model MicroFet 2 handheld dynamometer. Hand grip strength will be measured using a Jamar™ approved hydraulic dynamometer. The isometric strength of the hand and forearm muscles will be assessed with the arm extended along the body by performing a contraction for at least 3 seconds. The three maneuvers will be performed at least 20 seconds apart. The highest value obtained will be considered.

**Questionnaires.** The level of physical activity will be assessed using the RAPA [30] (Rapid Assessment of Physical Activity) questionnaire validated in Spanish population [31]. The scale was designed to measure this variable in older adults. It consists of 9 items, seven of which seek to determine whether people comply with the recommendation to perform 30 minutes or more of moderate physical activity at least 5 days a week. The two additional items measure whether people perform flexibility and strength exercises. Based on the total score, it establishes the following categories of physical activity: "Sedentary," "Not very active," "Not very active regular light," "Not very active regular" and "Active."

To assess QoL, the SF-12 questionnaire version 2 (SF-12v2), translated into Spanish and validated in Catalonian population [32], will be administered. It measures two main components: physical health (PCS) and mental health (MSC). It consists of 12 questions, with 8 dimensions: general self-perception of health, physical capacity, physical functioning, emotional role, social functioning, mental health, physical pain and self-perception of vitality with options. It measures

two main components, physical health (PCS) and mental health (MSC). The PCS and MCS scores are standardized on a scale from 0 to 100, where 50 is the mean for the general population. Higher scores indicate a better quality of life.

Anxiety and depression will be assessed using the Hospital Anxiety and Depression Scale (HADS), which is composed of 14 questions and divided into two subdimensions: anxiety (HADS-A) and depression (HADS-D) [33]. Each question has Likert-type responses, with scores ranging from 0 to 3. Thus, each subscale has a score ranging from 0 to 21 points. Scores between 0 and 7 indicate the absence of depression and/or anxiety, scores between 8 and 10 suggest a clinically significant disorder, and scores between 11 and 21 indicate the presence of moderate to severe depression and/or anxiety.

The Pittsburgh Sleep Quality Index (PSQI) [34] will be used to assess the participants' sleep quality. It is the most widely used self-report questionnaire for this purpose in the literature. It consists of 19 self-assessed questions whose overall rating ranges from 0 to 21. Higher scores represent poorer sleep quality. Above 5 is considered a "bad sleeper" and below 5 a "good sleeper." A change of 3 points is the minimum clinically significant difference.

The Duke scale [35] in its Spanish version [36] will be used to assess functional social support. It is a self-administered questionnaire with a Likert-type response scale (1–5). The score ranges from 11 to 55 points. The score obtained reflects perceived support rather than actual support. The lower score means less support. A score of 32 or more indicates widespread social support, whereas a score of less than 32 indicates low perceived social support.

To assess erectile dysfunction in male participants, the International Index of Erectile Function in its reduced version (IIEF-5) and validated in Spanish [37] will be used. It is a 5-question self-assessment questionnaire that addresses aspects such as confidence in achieving erections, rigidity of erections, ability to maintain an erection during sexual intercourse, and satisfaction with sexual activity. Each question is answered on a scale of 1–5, where 5 is the highest score and 1 is the lowest. The total score is obtained by adding the scores of each question, with a range of 5–25. It allows to classify erectile dysfunction as mild (17–21), moderate (12–16) or severe (5–11).

To determine female sexual dysfunction, participants will be given the Female Sexual Function Questionnaire [38]. This is a self-administered scale that assesses 10 domains of sexual activity. It consists of a total of 14 items, following a key question. Each domain comprises between 1 and 3 items. Each item is scored between 1 and 5. A score of 1 represents poor sexual function and a score of 5 represents normal or problem-free sexual function. The sum of the scores for the items in each domain is used to assess each domain.

To establish the severity of dyspnea, the scale created by the New York Heart Association (NYHA) [39] will be used. It reflects the patient's capacity for physical exertion and is therefore also called "NYHA functional class, which has four grades indicating greater dyspnea to a greater degree.

Eating habits will be evaluated using the Mediterranean diet adherence questionnaire [40]. It consists of 14 direct questions on the consumption of the main foods of the Mediterranean diet, such as olive oil, fruits, vegetables, legumes, fish, nuts, wine, and white meat. The scores are grouped into four categories: high adherence (12–14 points), medium adherence (8-11.99 points), low adherence (0-7.99 points).

**Body composition and nutritional measurements.** Bioimpedance will be used to determine the body composition of the participants. The measurement will be made with an Akern bioimpedance meter (Nutrilab model). The body mass index (BMI) of each participant will be calculated after weighing and measuring them in the training room. Waist circumference will also be measured at the level of the navel with a tape measure. Nutritional ultrasound [41] will also be performed. The FujiFilm Sonosite SII ultrasound scanner will be used. The subcutaneous fat in the quadriceps without contraction, thickness of the rectus femoris muscle (anteroposterior distance), transverse diameter, area in cm$^2$, and circumference will be measured. The thickness of the vastus intermedius will be measured (anteroposterior distance). In the abdomen without contraction, total subcutaneous fat, superficial subcutaneous fat, and pre-peritoneal fat will be measured.

## Sample size calculation

The sample size was calculated to detect differences between groups in our main outcome, CF, measured by $VO_2$max. According to the meta-analysis by Fabero-Garrido et al [16], people with ischemic heart disease who have undergone an IMT program have a mean difference of 2.18 ml $O_2$/kg/min with respect to the control group, and we want to find a deviation of 1 MET (3.5 ml $O_2$/kg/min). The calculations were performed using the statistical software G*Power 3.1, considering a confidence level of 95% and a power of 80% in all cases.

Based on the above, a sample size of 33 subjects per group (66 in total) was determined. In addition, to minimize the impact of possible dropouts, the total sample was increased by 10% to 72 patients, which will lead to the final inclusion of 36 patients in each group.

## Allocation concealment

Consecutive sampling will be performed by a rehabilitation physician on patients attending the CRU consultation of HUVV who meet the criteria (see *Eligibility criteria* section) and sign the informed consent. Once included, each patient will be randomly assigned to either the intervention or control group (1:1 ratio) using a random sequence of numbers generated with the Simple Random Sampling module of Epidat 3.1 (open-access epidemiological analysis software developed by the Xunta de Galicia, Dirección Xeral de Saúde Pública). Specifically, to reach a sample size of n = 72, the software will generate a sequence that determines the group for each participant according to their order of enrollment. This simple random assignment will ensure maximum unpredictability in the assignment of participants, preventing any prediction of future assignments. These assignments will be generated in advance by an independent team member who will develop sequentially numbered, truly opaque, and sealed envelopes. Finally, the rehabilitation physician responsible for recruitment will open the corresponding envelope to reveal which group the participant belongs to.

## Blinding

Participants will receive instruction about IMT from one of the rehabilitation physicians, who will be different from the one performing the assessments. Furthermore, none of the participants will know to which group they belong, since all of them will be given the same IMT device, but with a different training load. The physiotherapist and nurse who will conduct the CRP training sessions will not be involved in the allocation of participants. The statistical analysis will be carried out by a statistician of the research team who will not participate in the allocation, assessment or intervention process. In addition, the participants' data will be provided anonymously.

In the event of any adverse effect or circumstance that endangers the health of the participants, the researchers will be unblinded. If such a contingency arises, both a paper-based and an electronic record are always maintained and remain accessible to all authorized investigators.

## Data collection

Prior to inclusion in the study, each participant will undergo a simple ergometry test to determine their maximum heart rate (MHR), observe their behavior at maximum effort, and rule out conditions that contraindicate exercise. Besides, the rehabilitation physician will perform a nutritional assessment using bioimpedance analysis and a morphological assessment of the quadriceps and abdominal fat using ultrasonography. Patients will be weighed every week to monitor their evolution and rule out possible adverse effects of training. At the end of the program, ergometry, blood analysis, and the rest of the tests described below will be repeated.

**Status and timeline of the study.** Currently, patients are still being recruited for the study. This is planned to be completed on December 31, 2025. Data collection is expected to be concluded by March 31, 2026, and we expect to have the results ready for publication in the second quarter of 2026.

**Data management plan.** To ensure the privacy and confidentiality of participants, all clinical data collected in the study will be anonymized prior to analysis. Direct identifiers such as names, identification numbers, and specific dates will be removed, and indirect variables that could allow for the identification of subjects will be grouped and coded. The anonymization process will be carried out in accordance with current ethical and regulatory standards, ensuring that the data retained its validity for statistical analysis without compromising individual privacy. All physical documents (informed consent forms, data of the outcomes of the participants, etc.) will be stored in locked filing cabinets located in a secure area with restricted access. Electronic data will be entered into a secure, password-protected database designed specifically for the study. Only members of the research team listed in the protocol will be granted access to the data. The data that will support the findings of this study will not be openly available due to reasons of sensitivity and will be available from the corresponding author upon reasonable request. Data will be in controlled access data storage at University of Cadiz (Spain).

Participant data and variables will be gathered, as depicted in Figs 1 and 2.

## Statistical analysis

Descriptive statistics will summarize each variable, reporting means ± standard deviations (SD) or medians (IQR) for continuous data and absolute and relative frequencies for categorical data to verify baseline comparability between groups. Exploratory data analysis (e.g., box-plots, z-scores) will screen for influential outliers; values deemed spurious will be winsorized or, when justified, excluded, and sensitivity analyses will confirm that their treatment does not alter the main findings.

A mixed-design analysis of variance (ANOVA) will be performed to assess differential changes between groups over time (group × time interaction). The primary analysis strategy will depend on the proportion of missing data: if the dropout rate is less than 5%, a complete case analysis will be conducted, as this threshold is generally considered to have a

|  | STUDY PERIOD | | | |
|---|---|---|---|---|
|  | Enrolment | Allocation | Post-allocation | |
| **TIMEPOINT** | $t_{-1}$ | 0 | $t_0$ | $t_x$ |
| **ENROLMENT:** |  |  |  |  |
| Eligibility screen | X |  |  |  |
| Informed consent | X |  |  |  |
| Allocation |  | X |  |  |
| **INTERVENTIONS:** |  |  |  |  |
| *Experimental Group* CRP + IMT |  |  | ●———— | ————● |
| *Control Group* CRP |  |  | ●———— | ————● |
| $t_{-1}$, performed in a clinical setting; **0**, carried out prior to the 1st session; $t_0$, initial assessment and face-to-face contact; $t_x$, Post-intervention. *Abbreviations:* **CRP** cardiac rehabilitation program, **IMT** inspiratory muscle training. | | | | |

**Fig 1. Schedule of enrolment and interventions.**

 

| | STUDY PERIOD | | | |
|---|---|---|---|---|
| | **Enrolment** | **Allocation** | **Post-allocation** | |
| **TIMEPOINT** | $t_{-1}$ | **0** | $t_0$ | $t_x$ |
| **ASSESMENTS:** | | | | |
| **Baseline outcomes:** age, biological sex, marital status, education level, employment status, nationality and comorbidities. | | | X | |
| **Pre and post intervention variables:** | | | | |
| Functional capacity | | | X | X |
| Left ventricular ejection fraction | | | X | X |
| Maximal inspiratory pressure | | | X | X |
| Maximal expiratory pressure | | | X | X |
| Muscle strength | | | X | X |
| Body composition | | | X | X |
| Quality of life | | | X | X |
| Anxiety and depression | | | X | X |
| Level of physical activity | | | X | X |
| Dyspnea | | | X | X |
| Sleep quality | | | X | X |
| Social support | | | X | X |
| Erectile disfunction | | | X | X |
| Female sexual disfunction | | | X | X |
| Cardiological or non-cardiological incidences | | | ◆——————◆ | |
| Level of compliance | | | | X |

$t_{-1}$, performed in a clinical setting; **0**, carried out prior to the 1st session; $t_0$, initial assessment and face-to-face contact; $t_x$, Post-intervention.

**Fig 2. Schedule of assessments.**

negligible impact on bias. However, if missing data exceeds 5%, an intent-to-treat approach will be implemented, using multiple imputation to handle missing values and ensure that all randomized participants are included in the final model, in accordance with clinical trial standards [42].

All analyses will be performed in IBM SPSS Statistics v.24 (IBM Corp., Armonk, NY, USA), significance level will be set at 5% with 95% confidence intervals; assumptions, model fit, and potential risks (e.g., heteroscedasticity or multicollinearity) will be routinely checked, and predefined contingency strategies (nonparametric alternatives or robust estimators) will be implemented when violations are detected.

**Plan for detecting adverse effects and procedure to be followed.** During the entire procedure, all incidents, whether cardiological events or not, will be monitored and recorded. They will be analyzed by the researcher in charge of the study at the hospital and by the principal investigator, and if they are attributable to the intervention, the corresponding measures will be taken.

## Ethics

The present project will follow the ethical principles established for medical research on human subjects, according to the Declaration of Helsinki. The present clinical trial was approved by the Provincial Research Ethics Committee of Malaga (Spain) with Code SICEIA-2024–001869 in session number 9, held on 09/26/2024. Since data from real subjects will be used, it is essential to guarantee their confidentiality, privacy, and data protection. To this end, the participation of the subjects will be voluntary, and it will be an indispensable requirement to sign a written informed consent form for participation in the study. Participants will be able to withdraw from the study whenever they wish to do so. To safeguard the identity of the participants, their names and surnames will be coded in the database. This is a low-intervention clinical trial, since IMT is a very safe intervention, as it is routinely performed in other populations of chronic cardiac and respiratory patients without any relevant adverse effects having been reported. In addition, patients will be scrupulously stratified according to their cardiovascular risk, and the CRU at HUVV has extensive experience in training this type of patient, and its professionals are in constant training. The damages incurred by the participants will be covered by individual or collective self-insurance that the researcher in question has already contracted for the usual practice of his or her profession.

To publish the results of the project in international peer-reviewed journals and present the results at congresses worldwide, the Consolidated Standards of Reporting Trials (CONSORT) statement will be followed. In addition, as part of the knowledge translation strategy, we will disseminate the results on institutional websites and social media, contact news organizations, and identify partner institutions interested in the results. It is not planned to use professional scientific writers because the scientific articles will be written by the researchers of the study.

## Discussion

The present study will analyze the effect of adding IMT to a CRP in people with AMI who underwent revascularization by PCI. It is expected that CF and MIP will improve because there is some evidence (low and moderate) of this, as concluded in a recent review with meta-analysis by Fabero-Garrido et al. [16]. Furthermore, in similar clinical populations such as patients with HF, there is a high level of evidence that these outcomes and QoL, improve [18]. Therefore, it can be expected that the results are similar in AMI, in spite of no improvements in QoL have been evidenced yet in PCI patients. Nor have significant gains in sleep quality been observed to date, according to the available studies. In those variables without prior evidence, such as peripheral muscle strength, social support, anxiety, depression, coping with the disease, sexual dysfunction, eating habits, and body composition, we hypothesize that there will be an improvement, since increasing CF will reduce dyspnea during daily activities and thus could contribute, in our opinion, to the enhancement of other aspects of the individual. We decided to perform IMT at 70% of MIP based on a recent meta-analysis by Azambuja et al. [18]. As they concluded, training at loads above 60% of MIP had greater increases in inspiratory muscle strength, CF, and QoL in patients with HF. We expect these greater

effects to occur in people with AMI as well. In the case of CF, our main outcome, the improvement occurs for three reasons. First, early inspiratory muscle fatigue is delayed because it induces metaboloreflex which decreases blood flow to the peripheral musculature [43–46]. Second, blood supply to the peripheral musculature is increased by decreasing the blood demand of the inspiratory musculature [47–49]. Thirdly, IMT can reduce the peripheral chemoreflex response and enhance heart function. This is directly linked to decreased sympathetic activity in individuals with HF, leading to better systemic vasodilation, improved blood flow to peripheral muscles, greater ventilatory efficiency, and, as a result, increased CF [50]. Although undertaking a CRP itself improves psychosocial outcomes, dedicated IMT has further reduced fatigue, depression and perceived dyspnoea in patients with stable angina [51] and HF [52]. IMT has also demonstrated improvements in the psychosocial domain in healthy people [53] and people with chronic obstructive pulmonary disease [54,55]. Improved breathing efficiency may decrease the cognitive load associated with exertional breathlessness, fostering greater self-efficacy, exercise adherence and ultimately enhanced QoL. Women remain under-represented in cardiac rehabilitation (CR) trials, and it is a relevant aspect for the individualization of a CRP [56]. From a physiological perspective, women have lower respiratory muscle mass and lower basal MIP than men [57], which may predispose them to relatively greater improvements after IMT. Emerging evidence suggested that female patients obtain smaller absolute but similar relative gains from a CRP [58]. Therefore, our results will allow us to explore effect modification by sex, which may inform personalized prescriptions and help close the persistent gender gap in CR outcomes.

## Strengths and limitations

Our study has some strengths that need to be highlighted. The study is going to be carried out on a very specific population that has a specific pathology and has been treated in the same way, which makes the sample very homogeneous and therefore individuals very comparable with each other. Another strength is that this study will collect many variables, so we expect it to generate a lot of knowledge about this specific group of patients. This study is methodologically robust because it was prospectively registered and complies with the SPIRIT statement. In addition, it incorporates the Consensus on Exercise Reporting Template (CERT) as supporting information (S6 File) to include all the details of the rehabilitation and thus ensure that the exercise intervention can be replicated. It also performed concealed randomization and triple blinding. Additional strengths include a homogeneous, well-characterized sample and delivery within a real-world hospital CR setting supported by an experienced, multidisciplinary team. The primary outcome (VO$_2$ max) is objective, and all secondary outcomes will be measured with validated instruments, maximizing data quality and interpretability. We also believe that the long experience of the intervention team will contribute to the quality of this clinical trial. Finally, conducting the trial in a real-world hospital setting will facilitate rapid translation into routine cardiac-rehabilitation practice.

As limitations, we believe that the high number of questionnaires that the patients will have to answer could exhaust some of the participants and it could happen that they would not answer all of them. To try to avoid this, at the beginning of the intervention, the CRU nurse will explain to participants the importance of answering all the questionnaires. They will also be told that if they get tired, they can stop answering them and resume at another time of the day or on subsequent days. The fact that the trial is to be conducted in a real clinical setting with its possible errors and setbacks is also a limitation. Although standard operating procedures will be enforced, unexpected workflow disruptions could affect intervention fidelity. We are also aware that having the study subjects perform the IMT at home is not the best option even though we have provided all the necessary means (adequate instruction and performance record sheet) to make it as similar as possible to having it performed in the hospital (which is not possible due to organizational circumstances of the CRU). Because adherence relies on self-reported diaries and lacks electronic dose counters, compliance bias cannot be ruled out. No long-term follow-up beyond the 16 intervention sessions program means the durability of any observed benefits remains unknown.

 

## Conclusion

Current systematic reviews rate the evidence supporting IMT after PCI as having very low certainty, so guideline committees do not yet recommend its routine use in CRPs. This triple-blind, parallel-group, low-risk, randomized controlled trial aims to address this knowledge gap by evaluating the effectiveness of incorporating high-load IMT (70% maximal inspiratory pressure) into the standard 16 sessions of CRP in improving cardiorespiratory fitness, respiratory muscle strength, and a range of biopsychosocial outcomes in adults aged 18–80 with AMI who underwent revascularization by PCI.

If the study demonstrates clinically meaningful gains in functional capacity and related domains, the findings will support the incorporation of IMT as a core component of a CRP for this patient group. Conversely, null results will inform clinical practice by discouraging unnecessary interventions and guiding future multicenter trials. In either case, the trial will substantially advance the evidence base for optimizing rehabilitation after coronary revascularization.

## Supporting information

**S1 File. Spirit checklist.**
(PDF)

**S2 File. Cardioinspire protocol translated.**
(PDF)

**S3 File. Original Cardioinspire protocol.**
(PDF)

**S4 File. Consents translated.**
(PDF)

**S5 File. Original consents.**
(PDF)

**S6 File. CERT Cardiac rehabilitation program and CERT Inspiratory muscle Training.**
(PDF)

## Acknowledgments

We would like to thank the CRU of the Hospital Universitario Virgen de la Victoria (Málaga) and the University of Cadiz for their collaboration in the development of this study protocol.

## Author contributions

**Conceptualization:** Jose M. Zuazagoitia-Lama-Noriega, A.M. Gómez-González, Jose A. Moral-Munoz.

**Data curation:** Jose M. Zuazagoitia-Lama-Noriega.

**Formal analysis:** Jose M. Zuazagoitia-Lama-Noriega.

**Funding acquisition:** Jose M. Zuazagoitia-Lama-Noriega, A.M. Gómez-González.

**Investigation:** A.M. Gómez-González.

**Methodology:** Jose M. Zuazagoitia-Lama-Noriega, Jose A. Moral-Munoz.

**Project administration:** Jose A. Moral-Munoz.

**Writing – original draft:** Jose M. Zuazagoitia-Lama-Noriega.

**Writing – review & editing:** Jose M. Zuazagoitia-Lama-Noriega, A.M. Gómez-González, Jose A. Moral-Munoz.

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
