## [Decision Letter · Decision Letter 0]

19 Nov 2025

Dear Dr. Zuazagoitia-Lama-Noriega,

Thank you for submitting your manuscript to PLOS ONE. After careful consideration, we feel that it has merit but does not fully meet PLOS ONE’s publication criteria as it currently stands. Therefore, we invite you to submit a revised version of the manuscript that addresses the points raised during the review process.

We look forward to receiving your revised manuscript.

Kind regards,

Mansueto Gomes Neto, Ph.D

Academic Editor

PLOS ONE

Journal Requirements:

6. Please upload a copy of Supporting Information Figure S1, S2, which you refer to in your text on page 36.

Reviewers' comments:

Reviewer's Responses to Questions

**Comments to the Author**

1. Does the manuscript provide a valid rationale for the proposed study, with clearly identified and justified research questions?

Reviewer #1: Yes

Reviewer #2: Yes

2. Is the protocol technically sound and planned in a manner that will lead to a meaningful outcome and allow testing the stated hypotheses?

Reviewer #1: Yes

Reviewer #2: Partly

3. Is the methodology feasible and described in sufficient detail to allow the work to be replicable?

Reviewer #1: Yes

Reviewer #2: No

4. Have the authors described where all data underlying the findings will be made available when the study is complete?

Reviewer #1: Yes

Reviewer #2: Yes

5. Is the manuscript presented in an intelligible fashion and written in standard English?

Reviewer #1: Yes

Reviewer #2: Yes

You may also provide optional suggestions and comments to authors that they might find helpful in planning their study.

Reviewer #1: Interesting protocol.

Some issues should be added.

A primary end point should be clearly stated and sample size calculation performed on it

Abstract: in the conclusion I would not put numbers

Methods: probably more details about rehabilitation should be added in the appendix

methods: ways of blinding should be better detailed

methods: it should be added if block sizes were exploited

Reviewer #2: Comments to the Author

Overall, the protocol presents a relevant clinical question and outlines a study with potential to produce meaningful and applicable findings. However, as a study protocol, it is essential that all methodological components are described with complete transparency and precision to ensure reproducibility. Several elements—particularly the eligibility criteria, randomization procedures, allocation concealment, and statistical analysis plan—require clearer operational definitions and greater detail. Additionally, the rationale for specific intervention parameters—such as the choice of inspiratory muscle training at 70% of MIP—should be explicitly justified and supported by evidence to ensure appropriateness and feasibility. Strengthening these sections will enhance the robustness of the protocol and increase the reliability of the future trial.

Major Comment — Eligibility Criteria and Terminology (“Myocardial Ischemia”)

Page 6, line 121:

The term “myocardial ischemia (MI)” is potentially ambiguous and requires clearer definition. It is not evident whether the authors refer to individuals with acute myocardial infarction or to those with conditions involving reduced myocardial perfusion (e.g., stable coronary artery disease). Please specify the objective inclusion criteria (e.g., diagnostic biomarkers, ECG findings, imaging evidence) to ensure accurate characterization of the study population. Additionally, clarify whether the sample includes participants undergoing elective procedures, emergency procedures, or both. If a more precise descriptor is warranted, please apply the updated terminology consistently throughout the manuscript, including the title, abstract, introduction, and the “Participants and Study Setting” section.

Major Comment — Terminology: “Functional Capacity” vs. “Cardiorespiratory Fitness”

Page 9, lines 169–170:

In the Outcomes section, the term “functional capacity” is used to describe VO₂max obtained during maximal exercise testing. These measures are more accurately characterized as indicators of cardiorespiratory fitness rather than functional capacity in the clinical sense. I recommend replacing “functional capacity” with “cardiorespiratory fitness” in this section and ensuring that any earlier or subsequent uses of the term are revised for consistency and conceptual clarity.

Major Comment — Clarification of Revascularization Terminology (PTCA vs. PCI)

Page 6, line 109:

The terminology used to describe the revascularization procedure (“percutaneous transluminal coronary angioplasty – PTCA”) appears potentially inaccurate or incomplete. Contemporary cardiology literature typically uses the broader term percutaneous coronary intervention (PCI), which encompasses both balloon angioplasty (PTCA) and stent implantation. Because the manuscript title and several sections refer specifically to PTCA, it remains unclear whether the study will include only patients treated with balloon angioplasty, those treated with PCI involving stents, or both types of coronary interventions. This distinction should be clearly specified in the Eligibility Criteria section.

If the intention is to include all forms of PCI, the terminology should be updated consistently throughout the manuscript, including the title, abstract, introduction, and the Participants and Study Setting section, as well as all subsequent occurrences. Conversely, if the study is designed to include exclusively patients treated with PTCA, this should be explicitly stated and adequately justified, given that isolated PTCA is less common in contemporary clinical practice.

Major Comment — Adoption of the CONSORT, SPIRIT, and CERT Frameworks

In addition to the SPIRIT checklist for study protocols and the CONSORT guidelines for the forthcoming randomized clinical trial, I recommend incorporating the Consensus on Exercise Reporting Template (CERT) (doi:10.1136/bjsports-2016-096651). CERT will help ensure comprehensive reporting of exercise-related components in both the protocol and the final trial manuscript.

Major Comment — Allocation Concealment Not Adequately Ensured

The description of the allocation process suggests that group assignment may be predictable, which raises concerns regarding allocation concealment. Concealment must be ensured so that neither participants nor investigators involved in enrollment can foresee upcoming assignments. Please clarify the allocation procedure and describe the mechanism used to maintain adequate concealment (e.g., central randomization, sealed opaque envelopes). Notably, allocation concealment is conceptually distinct from blinding and must be addressed separately. This concern arises from the paragraph describing group allocation.

Major Comment — Clarity in Random Sequence Generation

The manuscript does not provide sufficient detail on how the random allocation sequence will be generated. Although Epidat 3.1 is mentioned, the specific method used (e.g., computer-generated random numbers, block randomization, stratification) is not described. Please specify how the software generates the random sequence, including any parameters used.

Major Comment — Statistical Analysis

The current statistical plan relies on multiple independent tests for within- and between-group comparisons. For a randomized trial with pre–post measurements, this approach does not adequately evaluate the group × time interaction. I recommend specifying a unified model (e.g., repeated-measures ANOVA or a linear mixed-effects model) to assess differential changes between groups over time. Additionally, the authors should explicitly state that analyses will follow the intent-to-treat principle and that the significance level will be set at 5% with 95% confidence intervals.

Minor Comments — Clarifications and Reference Updates

Page 4, line 79:

Endurance and strength training are mentioned, but no supporting reference is provided. Please include an appropriate citation.

Page 5, line 88:

The citation of the study by Fabero-Garrido may give the impression that the sample included only individuals undergoing PTCA. However, the original study included optimally treated individuals with ischemic heart disease, regardless of revascularization status. Please revise to ensure accurate representation.

Page 5, line 94:

The abbreviation “MIP” appears for the first time. Please define it as “maximal inspiratory pressure (MIP)” at first mention.

Minor Comment — Specification of Exercise Contraindications

Pages 6–7, lines 126–127:

The manuscript states that contraindications to exercise will be considered but does not list them. Please provide the specific contraindications or cite the guideline used to define them.

Minor Comment — Justification of the IMT Protocol Intensity

Page 8, lines 155–157: The inspiratory muscle training protocol (70% of MIP, 3 sets of 10 repetitions performing one daily training session four days a week) requires clearer justification. This intensity appears relatively high, and it is not clear whether this configuration has been tested in similar populations or shown to be effective and well tolerated. If supported by previous evidence, please cite the relevant studies. Otherwise, consider whether a more gradual progression of the training load—particularly during the initial sessions—might improve tolerance and adherence.

Minor Comment — Appropriateness of the NYHA Scale

Page 15, line 305: It is not clear whether all participants will have a diagnosis of heart failure. If not, the NYHA classification may not be the most appropriate tool for assessing dyspnea, as it is designed specifically for heart failure populations. For a broader cardiac sample, I recommend using an alternative validated instrument, such as the modified Medical Research Council scale (mMRC), which is applicable across different etiologies of dyspnea and may provide a more suitable assessment for this study.

**Do you want your identity to be public for this peer review?** For information about this choice, including consent withdrawal, please see our Privacy Policy

Reviewer #1: **Yes:** Fabrizio D'Ascenzo

Reviewer #2: No

---

## [Author Response · Author response to Decision Letter 1]

31 Dec 2025

REVIEWER #1

RV: Some issues should be added.

RV: 1. A primary end point should be clearly stated and sample size calculation

performed on it.

AA: We appreciate this observation and have clearly stated the primary endpoint, which

is “Cardiorespiratory Fitness” The sample size calculation was performed according to

the meta-analysis by Fabero-Garrido et al. (2024). This study established that people with

ischemic heart disease who have undergone an Inspiratory Muscle Training program have

a mean difference of 2.18 ml O2/kg/min with respect to the control group, and we want

to find a deviation of 1 MET (3.5 ml O2/kg/min). The calculations were performed using

the statistical software G*Power 3.1, considering a confidence level of 95% and power

of 80% in all cases.

- Fabero-Garrido R, Del Corral T, Plaza-Manzano G, Sanz-Ayan P, Izquierdo-

García J, López-De-Uralde-Villanueva I. Effects of Respiratory Muscle Training

on Exercise Capacity, Quality of Life, and Respiratory and Pulmonary Function

in People With Ischemic Heart Disease: Systematic Review and Meta-Analysis.

PTJ: Physical Therapy & Rehabilitation Journal | Physical Therapy [Internet].

2024;

RV: 2. Abstract: in the conclusion I would not put numbers.

AA: Thank you very much for your suggestion. We have reviewed the entire abstract

and have only left the number corresponding to the Clinical Trials registry, as this is a

requirement of the journal's guidelines.

RV: 3. Methods: probably more details about rehabilitation should be added in the

appendix.

AA: We have incorporated the Consensus on Exercise Reporting Template (CERT) as

supporting information (S10 File) to include all the details of the rehabilitation and

ensure that the exercise intervention can be replicated.

RV: 4. Methods: ways of blinding should be better detailed.

AA: In a specific section entitled “Blinding” (pages 19-20, lines 383-395), we have

adequately detailed the blinding of all parties involved: participants, rehabilitation

physicians, evaluators, physical therapist and nurse who will perform the intervention,

and statistician who will analyze the data. The following text was added to the main

document:

“Participants will receive instruction about IMT from one of the rehabilitation

physicians, who will be different from the one performing the assessments.

Furthermore, none of the participants will know to which group they belong, since all of

them will be given the same IMT device, but with a different training load.

The physiotherapist and nurse who will conduct the CRP training sessions will not be

involved in the allocation of participants.

The statistical analysis will be carried out by a statistician of the research team who

will not participate in the allocation, assessment or intervention process. In addition,

the participants’ data will be provided anonymously.

In the event of any adverse effect or circumstance that endangers the health of the

participants, the researchers will be unblinded. If such a contingency arises, both a

paper-based and an electronic record are always maintained and remain accessible to

all authorized investigators.”

RV: 5. Methods: it should be added if block sizes were exploited.

AA: We thank the reviewer for this important observation. We have revised and

significantly expanded the description of the random sequence generation process (page

18, lines 368-381). Each patient will be randomly assigned to either the intervention or

control group (1:1 ratio) using a random sequence of numbers generated with the

Simple Random Sampling module of Epidat 3.1 (open-access epidemiological analysis

software developed by the Xunta de Galicia, Dirección Xeral de Saúde Pública).

Specifically, to reach a sample size of N = 72, the software will generate random

numbers that will identify the participants. Participants will be randomly assigned to the

intervention group or control group based on the list of random numbers generated by

the software. Although block sizes will not be exploited, this simple random assignment

will ensure maximum unpredictability in the assignment of participants, preventing any

prediction of future assignments.

REVIEWER #2

Major Comments

RV: 1. Major Comment — Eligibility Criteria and Terminology (“Myocardial

Ischemia”). Page 6, line 121:

The term “myocardial ischemia (MI)” is potentially ambiguous and requires clearer

definition. It is not evident whether the authors refer to individuals with acute

myocardial infarction or to those with conditions involving reduced myocardial

perfusion (e.g., stable coronary artery disease). Please specify the objective inclusion

criteria (e.g., diagnostic biomarkers, ECG findings, imaging evidence) to ensure

accurate characterization of the study population. Additionally, clarify whether the

sample includes participants undergoing elective procedures, emergency procedures, or

both. If a more precise descriptor is warranted, please apply the updated terminology

consistently throughout the manuscript, including the title, abstract, introduction, and

the “Participants and Study Setting” section.

AA: We greatly appreciate your feedback. We have replaced the term “myocardial

ischemia” with “acute myocardial infarction” (code: BA41 from the International

Classification of Diseases, 11th revision), which specifically defines the study

population and, therefore, as you suggest, is more accurate. In addition, we have added

as an inclusion criterion (page 6, line 124) that an emergency percutaneous coronary

intervention has been performed. With these changes, it is clear that people with stable

angina or those who have undergone a scheduled procedure will not be included. We

have also updated all sections of the manuscript accordingly.

RV: 2. Major Comment — Terminology: “Functional Capacity” vs. “Cardiorespiratory

Fitness”

Page 9, lines 169–170: In the Outcomes section, the term “functional capacity” is used

to describe VO₂max obtained during maximal exercise testing. These measures are

more accurately characterized as indicators of cardiorespiratory fitness rather than

functional capacity in the clinical sense. I recommend replacing “functional capacity”

with “cardiorespiratory fitness” in this section and ensuring that any earlier or

subsequent uses of the term are revised for consistency and conceptual clarity.

AA: Thank you very much for your comment. Indeed, the term “cardiorespiratory

fitness” is more consistent, so we have replaced “functional capacity” with

“cardiorespiratory fitness”. We have also revised the term throughout the rest of the

document.

RV: 3. Major Comment — Clarification of Revascularization Terminology (PTCA vs.

PCI) Page 6, line 109.

The terminology used to describe the revascularization procedure (“percutaneous

transluminal coronary angioplasty – PTCA”) appears potentially inaccurate or

incomplete. Contemporary cardiology literature typically uses the broader term

percutaneous coronary intervention (PCI), which encompasses both balloon angioplasty

(PTCA) and stent implantation. Because the manuscript title and several sections refer

specifically to PTCA, it remains unclear whether the study will include only patients

treated with balloon angioplasty, those treated with PCI involving stents, or both types

of coronary interventions. This distinction should be clearly specified in the Eligibility

Criteria section.

If the intention is to include all forms of PCI, the terminology should be updated

consistently throughout the manuscript, including the title, abstract, introduction, and

the Participants and Study Setting section, as well as all subsequent occurrences.

Conversely, if the study is designed to include exclusively patients treated with PTCA,

this should be explicitly stated and adequately justified, given that isolated PTCA is less

common in contemporary clinical practice.

AA: We appreciate your suggestion and agree with it. We have changed the term

“percutaneous transluminal coronary angioplasty (PTCA)” to “percutaneous coronary

intervention (PCI)”. This clarifies the group of people we want to study which

encompasses both balloon angioplasty (PTCA) and stent implantation. The terminology

used in the manuscript in this regard has been modified accordingly.

RV: 4. Major Comment — Adoption of the CONSORT, SPIRIT, and CERT

Frameworks

In addition to the SPIRIT checklist for study protocols and the CONSORT guidelines

for the forthcoming randomized clinical trial, I recommend incorporating the Consensus

on Exercise Reporting Template (CERT) (doi:10.1136/bjsports-2016-096651). CERT

will help ensure comprehensive reporting of exercise-related components in both the

protocol and the final trial manuscript.

AA: We have incorporated the Consensus on Exercise Reporting Template (CERT) as

Supporting Information (S10 File) to ensure comprehensive reporting of exerciserelated

components.

RV: 5. Major Comment — Allocation Concealment Not Adequately Ensured

The description of the allocation process suggests that group assignment may be

predictable, which raises concerns regarding allocation concealment. Concealment must

be ensured so that neither participants nor investigators involved in enrollment can

foresee upcoming assignments. Please clarify the allocation procedure and describe the

mechanism used to maintain adequate concealment (e.g., central randomization, sealed

opaque envelopes). Notably, allocation concealment is conceptually distinct from

blinding and must be addressed separately. This concern arises from the paragraph

describing group allocation.

AA: We greatly appreciate your comments. We have rewritten the “Allocation

Concealment” section (pages 18-19, lines 367-381) and separated it from the “Blinding”

section (pages 19-20, lines 383-395) as you requested. We agree that a robust allocation

concealment mechanism is fundamental to preventing selection bias. To ensure that the

assignment is unpredictable and independent of the investigators involved in

recruitment, we will implement the following actions: a) The randomization sequence

will be generated by a team member who will not be involved in the clinical recruitment

or assessment of participants, using Epidat 3.1 software. b) This independent researcher

will prepare a set of identical, opaque, and carbon-lined envelopes. Each envelope will

be sequentially numbered on the outside and it will contain the group assignment on a

folded card inside. c) The envelopes will be opened by the rehabilitation physician only

after the patient meets all eligibility criteria and provides written informed consent. The

envelopes will be opened in strict numerical order, ensuring that the assignment for the

next participant will remain unknown until the previous one is fully enrolled.

RV: 6. Major Comment — Clarity in Random Sequence Generation.

The manuscript does not provide sufficient detail on how the random allocation

sequence will be generated. Although Epidat 3.1 is mentioned, the specific method used

(e.g., computer-generated random numbers, block randomization, stratification) is not

described. Please specify how the software generates the random sequence, including

any parameters used.

AA: Thank you very much again for the improvements you requested. Accordingly, the

following text was added (page 18, lines 370-377): “Once included, each patient will be

randomly assigned to either the intervention or control group (1:1 ratio) using a

random sequence of numbers generated with the Simple Random Sampling module of

Epidat 3.1 (open-access epidemiological analysis software developed by the Xunta de

Galicia, Dirección Xeral de Saúde Pública). Specifically, to reach a sample size of n =

72, the software will generate a sequence that determines the group for each participant

according to their order of enrollment. This simple random assignment will ensure

maximum unpredictability in the assignment of participants, preventing any prediction

of future assignments.”

RV: 7. Major Comment — Statistical Analysis

The current statistical plan relies on multiple independent tests for within- and betweengroup

comparisons. For a randomized trial with pre–post measurements, this approach

does not adequately evaluate the group × time interaction. I recommend specifying a

unified model (e.g., repeated-measures ANOVA or a linear mixed-effects model) to

assess differential changes between groups over time. Additionally, the authors should

explicitly state that analyses will follow the intent-to-treat principle and that the

significance level will be set at 5% with 95% confidence intervals.

AA: Thank you for your valuable comment. We have specified that the data will be

analyzed using a mixed-design ANOVA to assess differential changes between groups

over time. And we have also established that an intention-to-treat analysis with multiple

imputation will be implemented for missing data exceeding 5%; otherwise, a complete

case analysis will be conducted. Furthermore, we have specified that the significance

level will be set at 5% with 95% confidence intervals.

Minor Comments — Clarifications and Reference Updates

RV: 8. Page 4, line 79: Endurance and strength training are mentioned, but no

supporting reference is provided. Please include an appropriate citation.

AA: We have included the following reference that clearly highlights the obligation to

include strength training and aerobic exercise in cardiac rehabilitation programs.

- Brown TM, Pack QR, Aberegg E, Brewer LC, Ford YR, Forman DE, et al. Core

Components of Cardiac Rehabilitation Programs: 2024 Update: A Scientific

Statement From the American Heart Association and the American Association

of Cardiovascular and Pulmonary Rehabilitation. Circulation [Internet]. 2024

Oct 29;150(18):e328–47.

RV: 9. Page 5, line 88: The citation of the study by Fabero-Garrido may give the

impression that the sample included only individuals undergoing PTCA. However, the

original study included optimally treated individuals with ischemic heart disease,

regardless of revascularization status. Please revise to ensure accurate representation.

AA: Thank you very much for your comment. We have rewritten that sentence (page 5,

line 87), specifying that Fabero-Garrido's study includes patients with ischemic heart

disease so as not to give the impression that it only includes patients undergoing PTCA.

RV: 10. Page 5, line 94: The abbreviation “MIP” appears for the first time. Please

define it as “maximal inspiratory pressure (MIP)” at first mention.

AA: We appreciate you pointing that out, and we have defined “maximal inspiratory

pressure” for the first time.

Minor Comment — Specification of Exercise Contraindications

RV: Pages 6–7, lines 126–127: The manuscript states that contraindications to exercise

will be considered but does not list them. Please provide the specific contraindications

or cite the guideline used to define them.

AA: Thank you for your insightful comment. We have added the main specific

contraindications for starting cardiac rehabilitation in patients who have suffered an

acute myocardial infarction and have added the corresponding bibliographic reference:

- Protoview P, Vol B, Ringgold B. Guidelines for Cardiac Rehabilitation

Programs, 6th Edition. ProtoView [Internet]. 2020 [cited 2025 Dec

19];2020(19):1–2.

Minor Comment — Justification of the IMT Protocol Intensity

RV: Page 8, lines 155–157: The inspiratory muscle training protocol (70% of MIP, 3

sets of 10 repetitions performing one daily training session four days a week) requires

clearer justification. This intensity appears relatively high, and it is not clear whether

this configuration has been tested in similar populations or shown to be effective and

well tolerated. If supported by previous evidence, please cite the relevant studies.

Otherwise, consider whether a more gradual progression of the training load—

particularly during the initial sessions—might improve tolerance and adherence.

AA: The next three studies, which involved coronary artery bypass graft surgery and

coronary artery disease patients and used fairly similar IMT protocols in terms of sets,

repetitions, weekly sessions, and inten

---

## [Decision Letter · Decision Letter 1]

12 Feb 2026

Effectiveness of adding inspiratory muscle training to a cardiac rehabilitation program in people with acute myocardial infarction revascularized by percutaneous coronary intervention (CARDIOINSPIRE): Study protocol for a randomized controlled trial

PONE-D-25-40583R1

Dear Dr. Zuazagoitia-Lama-Noriega pleased to inform you that your manuscript has been judged scientifically suitable for publication and will be formally accepted for publication once it meets all outstanding technical requirements.

Kind regards,

Mansueto Gomes Neto, Ph.D

Academic Editor

PLOS One

Additional Editor Comments (optional):

Reviewers' comments:

Reviewer's Responses to Questions

**Comments to the Author**

1. Does the manuscript provide a valid rationale for the proposed study, with clearly identified and justified research questions?

Reviewer #1: Yes

2. Is the protocol technically sound and planned in a manner that will lead to a meaningful outcome and allow testing the stated hypotheses?

Reviewer #1: Yes

3. Is the methodology feasible and described in sufficient detail to allow the work to be replicable?

Reviewer #1: Yes

4. Have the authors described where all data underlying the findings will be made available when the study is complete?

Reviewer #1: Yes

5. Is the manuscript presented in an intelligible fashion and written in standard English?

Reviewer #1: Yes

You may also provide optional suggestions and comments to authors that they might find helpful in planning their study.

Reviewer #1: All comments have been addressed. authors should be complimented for performing a such complete review. I have no firther comments

**Do you want your identity to be public for this peer review?** For information about this choice, including consent withdrawal, please see our Privacy Policy

Reviewer #1: **Yes:** fabrizio d'ascenzo

---

## [Editor Report · Acceptance letter]

PONE-D-25-40583R1

PLOS One

Dear Dr. Zuazagoitia-Lama-Noriega,

I'm pleased to inform you that your manuscript has been deemed suitable for publication in PLOS One. Congratulations! Your manuscript is now being handed over to our production team.

Kind regards,

on behalf of

Dr. Mansueto Gomes Neto

Academic Editor

PLOS One